# The Modality Focusing Hypothesis: Towards Understanding Crossmodal Knowledge Distillation

**Zihui Xue**[*,1], **Zhengqi Gao**[*,2], **Sucheng Ren**[*,3], **Hang Zhao**[†,4]
[1] The University of Texas at Austin    [2] Massachusetts Institute of Technology
[3] South China University of Technology    [4] Tsinghua University, Shanghai Qi Zhi Institute

## Abstract

Crossmodal knowledge distillation (KD) extends traditional knowledge distillation to the area of multimodal learning and demonstrates great success in various applications. To achieve knowledge transfer across modalities, a pretrained network from one modality is adopted as the teacher to provide supervision signals to a student network learning from another modality. In contrast to the empirical success reported in prior works, the working mechanism of crossmodal KD remains a mystery. In this paper, we present a thorough understanding of crossmodal KD. We begin with two case studies and demonstrate that KD is not a universal cure in crossmodal knowledge transfer. We then present the modality Venn diagram (MVD) to understand modality relationships and the modality focusing hypothesis (MFH) revealing the decisive factor in the efficacy of crossmodal KD. Experimental results on 6 multimodal datasets help justify our hypothesis, diagnose failure cases, and point directions to improve crossmodal knowledge transfer in the future.[1]

## 1 Introduction

Knowledge distillation (KD) is an effective technique to transfer knowledge from one neural network to another (Wang & Yoon, 2021; Gou et al., 2021). Its core mechanism is a teacher-student learning framework, where the student network is trained to mimic the teacher through a loss. The loss function, initially proposed by (Hinton et al., 2015) as the KL divergence between teacher and student soft labels, has been extended in many ways (Zagoruyko & Komodakis, 2016; Tung & Mori, 2019; Park et al., 2019; Peng et al., 2019; Tian et al., 2019). KD has been successfully applied to various fields and demonstrates its high practical value.

The wide applicability of KD stems from its generality: *any* student can learn from *any* teacher. To be more precise, the student and teacher network may differ in several ways. Three common scenarios are: (1) *model capacity difference*: Many works (Zagoruyko & Komodakis, 2016; Tung & Mori, 2019; Park et al., 2019; Peng et al., 2019) on model compression aim to learn a lightweight student matching the performance of its cumbersome teacher for deployment benefits. (2) *architecture (inductive bias) difference*: As an example, recent works (Touvron et al., 2021; Ren et al., 2022; Xianing et al., 2022) propose to utilize a CNN teacher to distill its inductive bias to a transformer student for data efficiency. (3) *modality difference*: KD has been extended to transfer knowledge across modalities (Gupta et al., 2016; Aytar et al., 2016; Zhao et al., 2018; Garcia et al., 2018; Thoker & Gall, 2019; Ren et al., 2021; Afouras et al., 2020; Valverde et al., 2021; Xue et al., 2021), where the teacher and student network come from different modalities. Examples include using an RGB teacher to provide supervision signals to a student network taking depth images as input, and adopting an audio teacher to learn a visual student, etc.

Despite the great empirical success reported in prior works, the working mechanism of KD is still poorly understood (Gou et al., 2021). This puts the efficacy of KD into question: Is KD always

---

[*]Zihui, Zhengqi, and Sucheng contribute equally. Work is done during internship at Shanghai Qi Zhi Institute.
[†]Correspond to hangzhao@mail.tsinghua.edu.cn.

[1]Our code is available at https://github.com/zihuixue/MFH.

efficient? If not, what is a good indicator of KD performance? A few works (Cho & Hariharan, 2019; Tang et al., 2020; Ren et al., 2022) are in search for the answer in the context of *model capacity difference* and *architecture difference*. However, the analysis for the third scenario, KD under modality difference or formally *crossmodal KD*, remains an open problem. This work aims to fill this gap and for the first time provides a comprehensive analysis of crossmodal KD. Our major contributions are the following:

- We evaluate crossmodal KD on a few multimodal tasks and find surprisingly that teacher performance does not always positively correlate with student performance.
- To explore the cause of performance mismatch in crossmodal KD, we adopt the modality Venn diagram (MVD) to understand modality relationships and formally define *modality-general decisive features* and *modality-specific decisive features*.
- We present the modality focusing hypothesis (MFH) that provides an explanation of when crossmodal KD is effective. We hypothesize that *modality-general decisive features* are the crucial factor that determines the efficacy of crossmodal KD.
- We conduct experiments on 6 multimodal datasets (*i.e.*, synthetic Gaussian, AV-MNIST, RAVDESS, VGGSound, NYU Depth V2, and MM-IMDB). The results validate the proposed MFH and provide insights on how to improve crossmodal KD.

## 2    RELATED WORK

### 2.1    UNIMODAL KD

KD represents a general technique that transfers information learned by a teacher network to a student network, with applications to many vision tasks (Tung & Mori, 2019; Peng et al., 2019; He et al., 2019; Liu et al., 2019). Despite the development towards better distillation techniques or new application fields, there is limited literature (Phuong & Lampert, 2019; Cho & Hariharan, 2019; Tang et al., 2020; Ren et al., 2022; 2023) on understanding the working mechanism of KD. Specifically, Cho & Hariharan (2019) and Mirzadeh et al. (2020) investigate KD for model compression, *i.e.*, when the student and teacher differ in model size. They point out that mismatched capacity between student and teacher network can lead to failure of KD. Ren et al. (2022) analyze KD for vision transformers and demonstrate that teacher's inductive bias matters more than its accuracy in improving performance of the transformer student. These works provide good insight into understanding KD, yet their discussions are limited to unimodality and have not touched on KD for multimodal learning.

### 2.2    CROSSMODAL KD

With the accessibility of the Internet and the growing availability of multimodal sensors, multimodal learning has received increasing research attention (Baltrušaitis et al., 2018). Following this trend, KD has also been extended to achieve knowledge transfer from multimodal data and enjoys diverse applications, such as action recognition (Garcia et al., 2018; Luo et al., 2018; Thoker & Gall, 2019), lip reading (Ren et al., 2021; Afouras et al., 2020), and medical image segmentation (Hu et al., 2020; Li et al., 2020). Vision models are often adopted as teachers to provide supervision to student models of other modalities, *e.g.*, sound (Aytar et al., 2016; Xue et al., 2021), depth (Gupta et al., 2016; Xue et al., 2021), optical flow (Garcia et al., 2018), thermal (Kruthiventi et al., 2017), and wireless signals (Zhao et al., 2018). Although these works demonstrate potentials of crossmodal KD, they are often associated with a specific multimodal task. An in-depth analysis of crossmodal KD is notably lacking, which is the main focus of this paper.

### 2.3    MULTIMODAL DATA RELATIONS

There is continuous discussion on how to characterize multimodal (or multi-view) data relations. Many works (Tsai et al., 2020; Lin et al., 2021; 2022) utilize the multi-view assumption (Sridharan & Kakade, 2008), which states that either view alone is sufficient for the downstream tasks. However, as suggested in (Tsai et al., 2020), when the two views of input lie in different modalities, the multi-view assumption is likely to fail.[2] In the meantime, a few works on multimodal learning (Wang et al.,

---

[2]A detailed comparison of our proposed MVD with the multi-view assumption is presented in Appendix C.

2016; Zhang et al., 2018; Hazarika et al., 2020; Ma et al., 2020) indicate that multimodal features can be decomposed as modality-general features and specific features in each modality. Building upon these ideas, in this work, we present the MVD to formally characterize modality relations.

In addition, the importance of modality-general information has been identified in these works, yet with different contexts. In multi-view learning, (Lin et al., 2021; 2022) consider shared information between two views as the key to enforce cross-view consistency. To boost multimodal network performance and enhance its generalization ability, (Wang et al., 2016; Zhang et al., 2018; Hazarika et al., 2020; Ma et al., 2020) propose different ways to separate modality-general and modality-specific information. For semi-supervised multimodal learning, (Sun et al., 2020) aims at maximizing the mutual information shared by all modalities. To the best of our knowledge, our work is the first to reveal the importance of modality-general information in crossmodal KD.

## 3 ON THE EFFICACY OF CROSSMODAL KD

First, we revisit the basics of KD and introduce notations used throughout the paper. Consider a supervised $K$-class classification problem. Let $\mathbf{f}_{\boldsymbol{\theta}_s}(\mathbf{x}) \in \mathbb{R}^K$ and $\mathbf{f}_{\boldsymbol{\theta}_t}(\mathbf{x}) \in \mathbb{R}^K$ represent the output (*i.e.*, class probabilities) of the student and teacher networks respectively, where $\{\boldsymbol{\theta}_s, \boldsymbol{\theta}_t\}$ are learnable parameters. Without loss of generality, we limit our discussion within input data of two modalities, denoted by $\mathbf{x}^a$ and $\mathbf{x}^b$ for modality $a$ and $b$, respectively. Assume that we aim to learn a student network that takes $\mathbf{x}^b$ as input. In conventional *unimodal KD*, the teacher network takes input from the same modality as the student network (*i.e.*, $\mathbf{x}^b$). The objective for training the student is:

$$\mathcal{L} = \rho \mathcal{L}_{task} + (1 - \rho)\mathcal{L}_{kd} \tag{1}$$

where $\mathcal{L}_{task}$ represents the cross entropy loss between the ground truth label $y \in \{0, 1, \cdots, K-1\}$ and the student prediction $\mathbf{f}_{\boldsymbol{\theta}_s}(\mathbf{x}_b)$, $\mathcal{L}_{kd}$ represents the KL divergence between the student prediction $\mathbf{f}_{\boldsymbol{\theta}_s}(\mathbf{x}^b)$ and the teacher prediction $\mathbf{f}_{\boldsymbol{\theta}_t}(\mathbf{x}^b)$, and $\rho \in [0, 1]$ weighs the importance of two terms $\mathcal{L}_{task}$ and $\mathcal{L}_{kd}$ (*i.e.*, driving the student to true labels or teacher's soft predictions).

*Crossmodal KD* resorts to a teacher from the other modality (*i.e.*, $\mathbf{x}^a$) to transfer knowledge to the student. Eq. (1) is still valid with a slight correction that the KL divergence term is now calculated using $\mathbf{f}_{\boldsymbol{\theta}_s}(\mathbf{x}^b)$ and $\mathbf{f}_{\boldsymbol{\theta}_t}(\mathbf{x}^a)$. In addition, there is one variant (or special case) of crossmodal KD, where a multimodal teacher taking input from both modality $a$ and $b$ is adopted for distillation, and $\mathcal{L}_{kd}$ is now a KL divergence term between $\mathbf{f}_{\boldsymbol{\theta}_s}(\mathbf{x}^b)$ and $\mathbf{f}_{\boldsymbol{\theta}_t}(\mathbf{x}^a, \mathbf{x}^b)$.

We first present a case study on the comparison of crossmodal KD with unimodal KD. Consider the special case of crossmodal KD where a multimodal teacher is adopted. Intuitively, adopting a multimodal teacher, which takes both modality $a$ and $b$ as input, can be beneficial for distillation since: (1) a multimodal network usually enjoys a higher accuracy than its unimodal counterpart (Baltrušaitis et al., 2018), and a more accurate teacher ought to result in a better student; (2) the complementary modality-dependent information brought by a multimodal teacher can enrich the student with additional knowledge. This idea motivates many research works (Luo et al., 2018; Hu et al., 2020; Valverde et al., 2021) to replace a unimodal teacher with a multimodal one, in an attempt to improve student performance. Despite many empirical evidence reported in prior works, in this paper, we reflect on this assumption and ask the question: *Is crossmodal KD always effective?*

Table 1: Evaluation of unimodal KD (UM-KD) and crossmodal KD (CM-KD) on AV-MNIST and NYU Depth V2. 'Mod.' is short for modality, 'mIoU' denotes mean Intersection over Union, and A, I, RGB, D represents audio, grayscale images, RGB images and depth images, respectively.

| | AV-MNIST | | | | NYU Depth V2 | | | |
| --- | --- | --- | --- | --- | --- | --- | --- | --- |
| | Teacher | | Student | | Teacher | | Student | |
| | Mod. | Acc. (%) | Mod. | Acc. (%) | Mod. | mIoU (%) | Mod. | mIoU (%) |
| No-KD | - | - | A | 68.36 | - | - | RGB | 46.36 |
| UM-KD | A | 84.57 | A | 70.10 | RGB | 46.36 | RGB | 48.00 |
| CM-KD | I + A | 91.61 (↑) | A | 69.73 (↓) | RGB + D | 51.00 (↑) | RGB | 47.78 (↓) |

Table 1 provides two counterexamples for the above question on AV-MNIST and NYU Depth V2 data. The goal is to improve an audio student using KD on AV-MNIST and to improve an RGB model

on NYU Depth V2. From Table 1, we can see that a more accurate multimodal network does not serve as a better teacher in these two cases. For AV-MNIST, while the audio-visual teacher itself has a much higher accuracy than the unimodal teacher (*i.e.*, +7.04%), the resulting student is worse (*i.e.*, -0.37%) instead. Similarly, the great increase in teacher performance (*i.e.*, +4.64%) does not translate to student improvement (*i.e.*, -0.22%) for NYU Depth V2. These results cast doubt on the efficacy of crossmodal KD.[3] Even with the great increase in teacher accuracy, crossmodal KD fails to outperform unimodal KD in some cases. Contradictory to the previous intuition, teacher performance seems not reflective of student performance. Inspired by this observation, our work targets on exploring the open problem: *What is the fundamental factor deciding the efficacy of crossmodal KD?*

## 4 PROPOSED APPROACH

### 4.1 THE MODALITY VENN DIAGRAM

To study crossmodal KD, it is critical to first establish an understanding of multimodal data. Before touching multimodal data, let us fall back and consider unimodal data. Following a causal perspective (Schölkopf et al., 2012) (*i.e.*, features cause labels), we assume that the label $y$ is determined by a subset of features in $\mathbf{x}^a$ (or $\mathbf{x}^b$); this subset of features are referred to as *decisive features* for modality $a$ (or modality $b$) throughout the paper. For instance, colors of an image help identify some classes (*e.g.*, distinguish between a zebra and a horse) and can be considered as *decisive features*.

When considering multimodal data, input features of the two modalities will have logical relations such as intersection and union. We describe the modality Venn diagram (MVD) below to characterize this relationship. Stemming from the common perception that multimodal data possess shared information and preserve information specific to each modality, MVD states that any multimodal features are composed of *modality-general features* and *modality-specific features*. *Decisive features* of the two modalities are thus composed of two parts: (1) *modality-general decisive features* and (2) *modality-specific decisive features*; these two parts of *decisive features* work together and contribute to the final label $y$. Fig. 1 left shows an example of a video-audio data pair, where the camera only captures one person due to its position angle and the audio is mixed sounds of two instruments. Fig. 1 right illustrates how we interpret these three features (*i.e.*, *modality-general decisive*, visual *modality-specific decisive* and audio *modality-specific decisive*) at the input level.

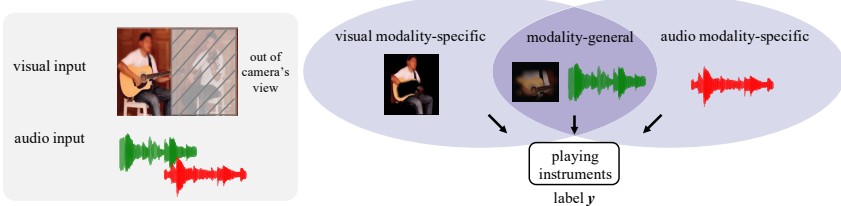

Figure 1: An input video-audio pair can be regarded as composed of *modality-general features* and *modality-specific features* in the visual and audio modality. For instance, the man playing violin on the right is not captured by the camera and hence its sound (marked in red) belongs to audio modality-specific information.

Next, we propose a formal description of MVD to capture the generating dynamics of multimodal data. Let $\mathcal{X}^a$, $\mathcal{X}^b$, and $\mathcal{Y}$ be the feature space of modality $a$, modality $b$, and the label space, respectively, and $(\mathbf{x}^a, \mathbf{x}^b, y)$ be a pair of data drawn from a unknown distribution $\mathcal{P}$ over the space $\mathcal{X}^a \times \mathcal{X}^b \times \mathcal{Y}$. MVD assumes that $(\mathbf{x}^a, \mathbf{x}^b, y)$ is generated by a quadruple $(\mathbf{z}^{sa}, \mathbf{z}^{sb}, \mathbf{z}^0, y) \in \mathcal{Z}^{sa} \times \mathcal{Z}^{sb} \times \mathcal{Z}^0 \times \mathcal{Y}$, following the generation rule:

$$\text{MVD GENERATION RULE:} \quad \mathbf{x}^a = \mathbf{g}^a(\mathbf{z}^a), \quad \mathbf{z}^a = [\mathbf{z}^{sa}, \mathbf{z}^0]^T \in \mathcal{Z}^a = \mathcal{Z}^{sa} \times \mathcal{Z}^0$$
$$\mathbf{x}^b = \mathbf{g}^b(\mathbf{z}^b), \quad \mathbf{z}^b = [\mathbf{z}^{sb}, \mathbf{z}^0]^T \in \mathcal{Z}^b = \mathcal{Z}^{sb} \times \mathcal{Z}^0 \quad (2)$$

where collectively, $\mathbf{g}^u(\cdot) : \mathcal{Z}^u \mapsto \mathcal{X}^u$ denotes an unknown generating function, if we adopt the notation $u \in \{a, b\}$. To complete the MVD, another linear decision rule should be included.

---

[3]Note that we even give preferable treatment to crossmodal KD: we take a multimodal network as teacher, and this teacher achieves higher accuracy than a teacher typically used in crossmodal KD.

Specifically, the following equation:

$$\text{MVD DECISION RULE:} \quad \exists \, \mathbf{W}^u, \; \arg\max\left[\text{Softmax}(\mathbf{W}^u\mathbf{z}^u)\right] = \arg\max\left[\mathbf{W}^u\mathbf{z}^u\right] = y \quad (3)$$

is assumed to hold for any $(\mathbf{z}^{sa}, \mathbf{z}^{sb}, \mathbf{z}^0, y)$, where we slightly abuse $\arg\max[\cdot]$ and here it means the index of the largest element in the argument. In essence, MVD specifies that $\mathbf{x}^u$ is generated based on a *modality-specific decisive feature* vector $\mathbf{z}^{su}$ and a *modality-general decisive feature* vector $\mathbf{z}^0$ (generation rule), and that $\mathbf{z}^u$ is sufficient to linearly determine the label $y$ (decision rule).

We proceed to quantify the correlation of *modality-general decisive features* and *modality-specific decisive features*. Let $\mathcal{Z}^{su} \subseteq \mathbb{R}^{d_{su}}$ and $\mathcal{Z}^0 \subseteq \mathbb{R}^{d_0}$, so that $\mathcal{Z}^u \subseteq \mathbb{R}^{d_u}$, where $d_u = d_{su} + d_0$. We denote a ratio $\gamma = d_0/(d_0 + d_{sa} + d_{sb}) \in [0,1]$, which characterizes the ratio of *modality-general decisive features* over all *decisive features*. Similarly, $\alpha = d_{sa}/(d_0 + d_{sa} + d_{sb})$ and $\beta = d_{sb}/(d_0 + d_{sa} + d_{sb})$ denotes the proportion of *modality-specific decisive features* for modality $a$ and $b$ over all *decisive features*, respectively, and we have $\alpha + \beta + \gamma = 1$.

### 4.2 THE MODALITY FOCUSING HYPOTHESIS

Based on MVD, we now revisit our observation in Sec. 3 (*i.e.*, teacher accuracy is not a key indicator of student performance) and provide explanations. First, teacher performance is decided by both *modality-general decisive* and *modality-specific decisive features* in modality $a$. In terms of student performance, although *modality-specific decisive features* in modality $a$ are meaningful for the teacher, they can not instruct the student since the student only sees modality $b$. On the other hand, *modality-general decisive features* are not specific to modality $b$ and could be transferred to the student. Coming back to the example in Fig. 1, if an audio teacher provides modality-specific information (*i.e.*, the sound colored in red), the visual student will get confused as this information (*i.e.*, playing violin) is not available in the visual modality. On the contrary, modality-general information can be well transferred across modalities and facilitates distillation as the audio teacher and visual student can both perceive the information about the left person playing guitar. This motivates the following modality focusing hypothesis (MFH).

**The Modality Focusing Hypothesis (MFH).** *For crossmodal KD, distillation performance is dependent on the proportion of modality-general decisive features preserved in the teacher network: with larger $\gamma$, the student network is expected to perform better.*

The hypothesis states that in crossmodal knowledge transfer, the student learns to "focus on" *modality-general decisive features*. Crossmodal KD is thus beneficial for the case where $\gamma$ is large (*i.e.*, multimodal data share many label-relevant information). Moreover, it accounts for our observation that teacher performance fails to correlate with student performance in some scenarios — When $\alpha$ is large and $\gamma$ is small, the teacher network attains high accuracy primarily based on modality-specific information, which is not beneficial for the student's learning process.

To have an intuitive and quick understanding of our hypothesis, here we present two experiments with synthetic Gaussian data. More details can be found in Sec. 5.2. As shown in Fig. 2, we start from the extreme case where two modalities do not overlap, and gradually increase the proportion of *modality-general decisive features* until all *decisive features* are shared by two modalities. We observe that crossmodal KD fails to work when $\mathbf{x}^a$ and $\mathbf{x}^b$ share few *decisive features* (*i.e.*, $\gamma$ is small) since *modality-specific decisive features* in modality $a$ are not perceived by the student. As

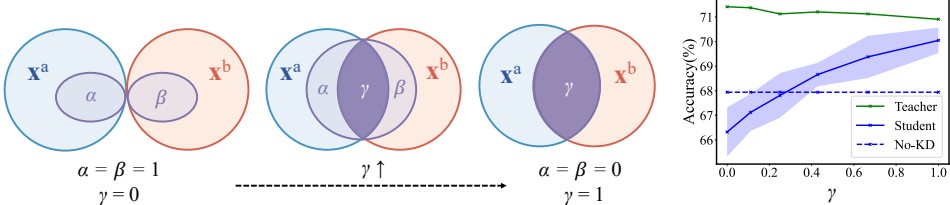

Figure 2: An illustration of MFH with synthetic Gaussian data. Teacher modality is $\mathbf{x}^a$ and student modality is $\mathbf{x}^b$. We plot the confidence interval of one standard deviation for student accuracy. With increasing $\gamma$, crossmodal KD becomes more effective.

Figure 3: With increasing $\alpha$ (*i.e.*, decreasing $\gamma$), the teacher improves its prediction accuracy but the student network fails to benefit from KD. See the caption of Fig. 2 for more explanations.

$\gamma$ gradually increases, crossmodal KD becomes more effective. For the case where all *decisive features* possess in both modalities, the student gains from teacher's knowledge and outperforms its baseline by 2.1%. Note that the teacher accuracy does not vary much during this process, yet student performance differs greatly.

Fig. 3 illustrates the reverse process where *modality-specific decisive features* in modality $a$ gradually dominate. With increasing $\alpha$, the teacher gradually improves since it receives more *modality-specific decisive features* for prediction. However, the student network fails to benefit from the improved teacher and performs slightly worse instead. Clearly, teacher performance is not reflective of student performance in this case. These two sets of experiments help demonstrate that teacher accuracy does not faithfully reflect the effectiveness of crossmodal KD and lend support to our proposed hypothesis.

Apart from the two intuitive examples, below we provide a theoretical guarantee of MFH in an analytically tractable case, linear binary classification. Formally, considering an infinitesimal learning rate which turns the training into a continuous gradient flow defined on a time parameter $t \in [0, +\infty)$ (Phuong & Lampert, 2019). If $n$ data are available, which are collectively denoted as $\mathbf{Z}^u \in \mathbb{R}^{d_u \times n}$, we have the following theorem to bound the training distillation loss with $\gamma$.

**Theorem 1.** *(Crossmodal KD in linear binary classification). Without loss of generality, we assume $\mathbf{f}_{\boldsymbol{\theta}_t}(\cdot) : \mathcal{X}^a \mapsto \mathcal{Y}$ and $\mathbf{f}_{\boldsymbol{\theta}_s}(\cdot) : \mathcal{X}^b \mapsto \mathcal{Y}$. Suppose $\max\{||\mathbf{Z}^u\mathbf{Z}^{u,T}||, ||(\mathbf{Z}^u\mathbf{Z}^{u,T})^{-1}||\} \leq \lambda$ always holds for both $u = a$ or $b$, and $\mathbf{g}^u(\cdot)$ are identity functions. If there exists $(\epsilon, \delta)$ such that*

$$Pr\left[||\mathbf{Z}^{a,T}\mathbf{Z}^a - \mathbf{Z}^{b,T}\mathbf{Z}^b|| \leq (1-\gamma)\epsilon\right] \geq 1 - \delta \tag{4}$$

*Then, with an initialization at $t = 0$ satisfying $R_n^{dis}(\boldsymbol{\theta}_s(0)) \leq q$, we have, at least probability $1 - \delta$:*

$$R_n^{dis}(\boldsymbol{\theta}_s(t = +\infty)) \leq n\left(\frac{\epsilon^\star}{1 - e^{-\epsilon^\star}} - 1 - \ln\frac{\epsilon^\star}{1 - e^{-\epsilon^\star}}\right) \tag{5}$$

*where $\epsilon^\star = \lambda^{1.5}(\lambda^2 + 1)(1 - \gamma)\epsilon$ and $R_n^{dis}(\boldsymbol{\theta}_s)$ is the empirical risk defined by KL divergence (corresponding to Eq. (1) when $\rho = 0$):*

$$R_n^{dis}(\boldsymbol{\theta}_s(t)) = \sum_{i=1}^{n} -\sigma(\boldsymbol{\theta}_t^T\mathbf{x}_i^a) \cdot \ln\frac{\sigma(\boldsymbol{\theta}_s^T\mathbf{x}_i^b)}{\sigma(\boldsymbol{\theta}_t^T\mathbf{x}_i^a)} - \left[1 - \sigma(\boldsymbol{\theta}_t^T\mathbf{x}_i^a)\right] \cdot \ln\frac{1 - \sigma(\boldsymbol{\theta}_s^T\mathbf{x}_i^b)}{1 - \sigma(\boldsymbol{\theta}_t^T\mathbf{x}_i^a)} \tag{6}$$

See Appendix A for the omitting proof, several important remarks, and future improvements.

## 4.3 IMPLICATIONS

We have presented MVD and MFH. Equipped with this new perspective of crossmodal KD, we discuss their implications and practical applications in this section.

**Implication.** For crossmodal KD, consider two teachers with identical architectures and similar performance: Teacher (a) makes predictions primarily based on *modality-general decisive features* while teacher (b) relies more on *modality-specific decisive features*. We expect that the student taught by teacher (a) yields better performance than that by teacher (b).

The implication above provides us with ways to validate MFH. It also points directions to improve crossmodal KD — We can train a teacher network that focuses more on *modality-general decisive*

*features* for prediction. Compared with a regularly-trained teacher, the new teacher is more modality-general (*i.e.*, has a larger $\gamma$) and thus tailored for crossmodal knowledge transfer.

Note that: Firstly, identical architectures and performance of the two teachers are stated here for a fair comparison. Similar performance of the two teachers translate to similar ability to extract *decisive features* for prediction, and thus the only difference lies in the amount of *modality-general decisive features*. This design excludes other factors and helps justify that the performance difference stems from $\gamma$. In fact, we observe that even with inferior accuracy than teacher (b), teacher (a) still demonstrates better crossmodal KD performance in experiments. Secondly, the main focus of this paper is to present the MFH and to validate it with theoretical analysis, synthetic experiments, and evidence from experiments conducted on real-world multimodal data. Contrary to the common belief, we detach the influence of teacher performance in crossmodal KD and point out that *modality-general decisive features* are the key. Developing methods to separate *modality-general/specific decisive features* from real-world multimodal data is beyond the scope of this paper and left as future work.

## 5 EXPERIMENTAL RESULTS

### 5.1 EXPERIMENTAL SETUP

To justify our MFH, we conduct experiments on 6 multimodal datasets (synthetic Gaussian, AV-MNIST, RAVDESS, VGGSound, NYU Depth V2, and MM-IMDB) that cover a diverse combination of modalities including images, video, audio and text. In essence, we design approaches to obtain teachers of different $\gamma$ and perform crossmodal KD to validate the implication presented in Sec. 4.3.

We consider four different ways to derive a teacher network that attends to more or fewer *modality-general decisive features* than a regularly-trained teacher: (1) For synthetic Gaussian data, since the multimodal data generation mechanism is known, we train a modality-general teacher on data with only *modality-general decisive features* preserved and other channels removed; (2) For NYU Depth V2, we notice that RGB images and depth images share inherent similarities and possess identical dimensions, allowing them to be processed using a single network. Therefore, we design a modality-general teacher (*i.e.*, has a larger $\gamma$ than a regularly-trained teacher) by following the training approach in (Girdhar et al., 2022); (3) For MM-IMDB data, we follow the approach in (Xue et al., 2021) to obtain a multimodal teacher which is more modality-specific (*i.e.*, has a smaller $\gamma$) than the regularly-trained teacher; (4) For the other datasets, we design an approach based on feature importance (Breiman, 2001; Wojtas & Chen, 2020) to rank all feature channels according to the amount of modality-general decisive information. With a sorted list of all features, we train a modality-general teacher by only keeping feature channels with large salience values and a modality-specific teacher by keeping features with small values. In summary, a wide range of approaches and tasks are considered to justify MFH. See Appendix B for detailed setups and some results.

### 5.2 SYNTHETIC GAUSSIAN

Table 2: Results on synthetic Gaussian data. Compared with a regular teacher, the modality-general teacher has downgraded performance yet leads to a student with increasing accuracy.

| Regular Teacher | | Modality-general Teacher | | Student Acc. (%) | | |
|---|---|---|---|---|---|---|
| $\gamma$ | Acc. (%) | $\gamma$ | Acc. (%) | No-KD | Regular-KD | Modality-general-KD |
| 0.25 | 89.70 | 1.0 | 64.84 (↓) | 59.29 | 59.46 | 61.01 (↑) |
| 0.50 | 89.62 | 1.0 | 73.41 (↓) | 67.64 | 67.86 | 70.25 (↑) |
| 0.75 | 89.70 | 1.0 | 79.41 (↓) | 76.46 | 76.53 | 77.70 (↑) |

We extend the Gaussian example in (Lopez-Paz et al., 2015) to a multimodal scenario (see Appendix B.2 for details). To validate MFH, we train two teacher networks: (1) a regular teacher is trained on data with all input feature channels; (2) a modality-general teacher is trained on data with *modality-general decisive features* preserved and other channels removed, thus this teacher has $\gamma = 1$. We experiment with different data generation methods, and the regular teachers have different $\gamma$ correspondingly. Crossmodal KD results obtained by these two teachers are presented in Table 2. We observe considerable performance degradation (larger than -10% accuracy loss) of the modality-

general teacher than the regular teacher, as it only relies on *modality-general decisive features* and discards *modality-specific decisive features* for prediction. However, the modality-general teacher still facilitates crossmodal KD and leads to an improved student ($\sim$ +2% accuracy improvement compared with regular crossmodal KD). The results align well with MFH stating that a teacher with more emphasis on *modality-general decisive features* (*i.e.*, has a larger $\gamma$) yields a better student.

## 5.3 NYU DEPTH V2

We revisit the example of NYU Depth V2 in Sec. 3. We adopt a teacher network that takes depth images as input to transfer knowledge to an RGB student. Both student and teacher network architectures are implemented as DeepLab V3+ (Chen et al., 2018). As described in Sec. 5.1, besides training a regular teacher, we follow (Girdhar et al., 2022) and train a teacher that learns to predict labels for the two modalities with identical parameters. To be specific, a training batch contains both RGB and depth images, and the teacher network is trained to output predictions given either RGB or depth images as input. As such, the resulting teacher is assumed to extract more modality-general features for decision (*i.e.*, has a larger $\gamma$ than a regular teacher) since it needs to process both modalities in an identical way during training.

As shown in Table 3, regular crossmodal KD does not bring many advantages: the student achieves a similar mIoU compared with the No-KD baseline. Therefore, one might easily blame the failure of crossmodal KD on teacher accuracy and assume that crossmodal KD is not effective because the depth teacher itself yields poor performance (*i.e.*, has an mIoU of 37.33%). By nature of its training approach, the modality-general teacher is forced to extract more *modality-general decisive features* for prediction rather than rely on depth-specific features as it also takes RGB images as input. While we do not observe difference in teacher performance, the modality-general teacher turns out to be a better choice for crossmodal KD: its student mIoU improves from 46.36% to 47.93%. The results indicate that our MFH has the potential to diagnose crossmodal KD failures and lead to improvement.

Table 3: Results on NYU Depth V2 semantic segmentation. Teacher modality is depth images and student modality is RGB images. Compared with regular crossmodal KD, using a modality-general teacher leads to better distillation performance.

|  | Teacher | | Student | |
|---|---|---|---|---|
|  | Modality | mIoU (%) | Modality | mIoU (%) |
| No-KD | - | - | RGB | 46.36 |
| Regular-KD | D | 37.33 | RGB | 46.89 |
| Modality-general-KD | RGB/D | 37.47 | RGB | 47.93 (↑) |

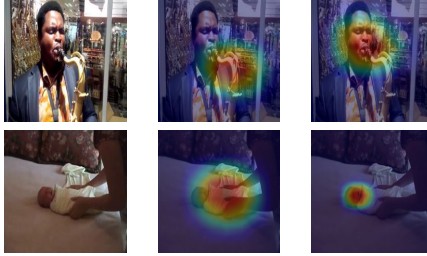

Figure 4: The class activation maps of a regular teacher (middle) and a modality-general teacher (right) on VGGSound. A regular teacher attends to all *decisive features* (*i.e.*, the visual objects) while a modality-general one focuses on *modality-general decisive features* (*i.e.*, the area of vocalization).

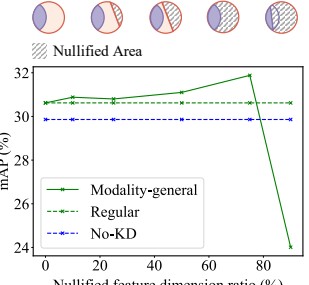

Figure 5: With more feature channels getting nullified, student performance starts to increase in the beginning and then decrease; the process aligns well with MVD.

## 5.4 RAVDESS AND VGGSOUND

Besides the approaches presented above, we design a permutation-based method to sort the features according to the salience of *modality-general decisive features*, which allows us to obtain teacher

networks that possess different amount of $\gamma$. See Appendix B.4 for the detailed algorithm flow. We apply this approach to RAVDESS (Livingstone & Russo, 2018) and VGGSound (Chen et al., 2020a). RAVDESS is an audio-visual dataset containing 1,440 emotional utterances with 8 different emotion classes. Teacher modality is audio and student modality is images. The student and teacher network follow the unimodal network design in (Xue et al., 2021); VGGSound is a large-scale audio-visual event classification dataset including over 200,000 video and 310 classes. We consider two setups: (1) adopting an audio teacher and a video student; (2) using a video teacher for distillation to an audio student. We also experiment with two architectures, ResNet-18 and ResNet-50 (He et al., 2016) to be the teacher and student network backbone. In our algorithm, we sort feature channels according to

Table 4: Results on RAVDESS Emotion Recognition and VGGSound Event Classification. `A`, `I` and `V` denotes audio, images and video, respectively. We report student test accuracy (%) for RAVDESS and mean Average Precision (%) for VGGSound. With identical model architecture, a modality-general teacher improves regular KD while a modality-specific teacher leads to downgraded performance.

| Task | T Mod. | S Mod. | Regular | Mod.-specific | Random | Mod.-general |
|------|--------|--------|---------|---------------|--------|--------------|
| RAVDESS | A | I | 77.22 | 42.66 (↓) | 64.75 | 78.28 (↑) |
| VGGSound (RN-18) | A | V | 30.62 | 24.98 (↓) | 28.99 | 31.88 (↑) |
| VGGSound (RN-50) | A | V | 38.78 | 28.65 (↓) | 35.95 | 39.81 (↑) |
| VGGSound (RN-50) | V | A | 58.87 | 31.28 (↓) | 56.98 | 59.46 (↑) |

the salience of *modality-general decisive features*. A modality-general teacher is derived by nullifying $r\%$ feature channels with smallest salience values. Similarly, a modality-specific teacher is obtained by nullifying the $r\%$ feature channels with largest salience values and a random teacher is derived by randomly nullifying $r\%$ channels. The feature nullifying ratio $r\%$ is a hyperparameter and we experiment with different values of $r$. While the exact value of $\gamma$ for different teacher networks is unknown, we know that $\gamma$ increases in the order of modality-specific, random and modality-general teacher. As shown in Table 4, crossmodal KD performance aligns well with this order. For RAVDESS emotion recognition, a modality-specific teacher results in a significantly downgraded student (*i.e.*, accuracy drops from 77.22% to 42.66%) while a modality-general teacher improves regular KD by 1.06%. Similarly, for VGGSound event classification, modality-general crossmodal KD improves the video student performance from 30.62% to 31.88% (for ResNet-18) and from 38.78% to 39.81% (for ResNet-50). These results demonstrate the critical role of *modality-general decisive features* in crossmodal KD.

Moreover, to provide an intuitive understanding of how a modality-general teacher differs from the regular teacher, we present visualization results on VGGSound data in Fig. 4. We see that a regular video teacher utilizes all *decisive features* for classification, and attends to the visual objects (*i.e.*, the Saxophone and the baby). On the contrary, a modality-general teacher focuses more on information available in both the visual and audio modality, thus the area of vocalization get most activated.

Finally, we vary feature nullifying ratio $r\%$ for VGGSound data and plot the student performance curve along with $r\%$. From Fig. 5, we observe that there exists a sweet spot for modality-general KD. As $r\%$ increases, the student performance improves in the beginning. The improvement indicates that *non modality-general decisive features* in the teacher are gradually discarded, which in turn results in a better student. Later, after all *non modality-general decisive* information are discarded, the feature nullifying process starts to hinder student performance as *modality-general decisive features* get nullified as well. The modality Venn diagram corresponding to this process is depicted in the upper figure. The observed performance curve aligns well with our understanding on MVD.

## 6 CONCLUSION AND FUTURE WORK

In this work, we present a thorough investigation of crossmodal KD. The proposed MVD and MFH characterize multimodal data relationships and reveal that *modality-general decisive features* are the key in crossmodal KD. We present theoretical analysis and conduct various experiments to justify MFH. We hope MFH shed light on applications of crossmodal KD and will raise interest for general understanding of multimodal learning as well. Future work includes: (i) deriving a more profound theoretical analysis of crossmodal KD, (ii) differentiating *modality-general/specific decisive* features for real-world data, and (iii) improving multimodal fusion robustness based on MVD.

ACKNOWLEDGEMENT

The authors would like to thank Hangyu Lin (HKUST) and Zikai Xiong (MIT) for providing crucial steps in proving Theorem 1, and the anonymous reviewers for valuable feedbacks.

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

## A  PROOF OF THEOREM 1

**Lemma 1.** *Consider a function $l(b, a)$:*

$$l(b, a) = -\sigma(a) \left[\ln \sigma(b) - \ln \sigma(a)\right] - (1 - \sigma(a)) \left[\ln(1 - \sigma(b)) - \ln(1 - \sigma(a))\right] \tag{7}$$

*where $\sigma(\cdot)$ represents the sigmoid function. When $|a - b| \leq \epsilon$,*

$$\max l(b, a) = \frac{\epsilon}{1 - e^{-\epsilon}} - 1 - \ln \frac{\epsilon}{1 - e^{-\epsilon}} \tag{8}$$

*Proof.* We only need to prove the following two things: (i) $l(b, a) \leq l(a \pm \epsilon, a)$, and (ii) $l(a \pm \epsilon, a) \leq \frac{\epsilon}{1 - e^{-\epsilon}} - 1 - \ln \frac{\epsilon}{1 - e^{-\epsilon}}$. For (i), we imagine $a$ is fixed, and calculate the derivative of $l(b, a)$ w.r.t. $b$, yielding:

$$\frac{\partial l}{\partial b} = \frac{e^b - e^a}{(1 + e^a)(1 + e^b)} = \sigma(b) - \sigma(a) \tag{9}$$

Thus, when varying $b$ in the range $[a - \epsilon, a + \epsilon]$, the value of $l(b, a)$ first decreases and then increases. It implies $l(b, a) \leq l(a \pm \epsilon, a)$.

To prove (ii), we first consider the function:

$$
\begin{aligned}
\tilde{l}_+(a) &= l(a + \epsilon, a) \\
&= -\sigma(a) \left[\ln \sigma(a + \epsilon) - \ln \sigma(a)\right] - (1 - \sigma(a)) \left[\ln(1 - \sigma(a + \epsilon)) - \ln(1 - \sigma(a))\right] \\
&= \sigma(a) \left[\ln \frac{\sigma(a)}{1 - \sigma(a)} - \ln \frac{\sigma(a + \epsilon)}{1 - \sigma(a + \epsilon)}\right] - \ln \frac{1 - \sigma(a + \epsilon)}{1 - \sigma(a)} \\
&= -\sigma(a) \cdot \epsilon - \ln \frac{1 - \sigma(a + \epsilon)}{1 - \sigma(a)} \\
&= -\epsilon \frac{1}{1 + e^{-a}} - \ln \frac{e^{-(a+\epsilon)}}{e^{-a}} \frac{1 + e^{-a}}{1 + e^{-(a+\epsilon)}} \\
&= -\epsilon \frac{1}{1 + e^{-a}} + \epsilon - \ln \frac{1 + e^{-a}}{1 + e^{-(a+\epsilon)}} \\
&= -\epsilon \frac{1}{1 + t} + \epsilon - \ln \frac{1 + t}{1 + te^{-\epsilon}}
\end{aligned}
\tag{10}
$$

where in the last line, we denote $t = e^{-a}$. Now, let us calculate the derivative of $\tilde{l}_+$ w.r.t. $t$:

$$
\begin{aligned}
\frac{\partial \tilde{l}_+}{\partial t} &= \frac{\epsilon}{(1 + t)^2} - \frac{1}{1 + t} + \frac{e^{-\epsilon}}{1 + te^{-\epsilon}} \\
&= \frac{\epsilon - 1 - t}{(1 + t)^2} + \frac{1}{e^\epsilon + t} \\
&= \frac{(1 + t)^2 - (t + e^\epsilon)(t - \epsilon + 1)}{(1 + t)^2(e^\epsilon + t)}
\end{aligned}
\tag{11}
$$

Obviously, the denominator is always larger than zero. After expanding the numerator, we see that when $t = \frac{e^\epsilon(1-\epsilon)-1}{1+\epsilon-e^\epsilon}$, $\tilde{l}_+$ achieves its maximum. Therefore, according to $a = -\ln t$, we conclude the maximum value of $\tilde{l}_+(a)$ is $\frac{\epsilon}{1 - e^{-\epsilon}} - 1 - \ln \frac{\epsilon}{1 - e^{-\epsilon}}$, achieved at $a = -\ln \frac{e^\epsilon(1-\epsilon)-1}{1+\epsilon-e^\epsilon}$. Similarly, we could define a function $\tilde{l}_-(a) = l(a - \epsilon, a)$ and verify that the maximum value of $\tilde{l}_-(a)$ is $\frac{-\epsilon}{1 - e^\epsilon} - 1 - \ln \frac{-\epsilon}{1 - e^\epsilon}$. Furthermore, after some simplifications, these two expressions, i.e., $\max \tilde{l}_+(a)$ and $\max \tilde{l}_-(a)$, could be proved identical. □

**Corollary 1.** *For two vectors $\mathbf{a} \in \mathbb{R}^n$ and $\mathbf{b} \in \mathbb{R}^n$, if $\|\mathbf{a} - \mathbf{b}\| \leq \epsilon$, then*

$$L_n(\mathbf{b}, \mathbf{a}) = \sum_{i=1}^n l(b_i, a_i) \leq n \left(\frac{\epsilon}{1 - e^{-\epsilon}} - 1 - \ln \frac{\epsilon}{1 - e^{-\epsilon}}\right) \tag{12}$$

The corollary is straightforward if noticing $|a_i - b_i| \leq ||\mathbf{a} - \mathbf{b}|| \leq \epsilon$ holds for any $i = 1, 2, \cdots, n$.

**Lemma 2.** *If* $\max\{||\mathbf{Z}^u\mathbf{Z}^{u,T}||, ||(\mathbf{Z}^u\mathbf{Z}^{u,T})^{-1}||\} \leq \lambda$ *always holds for both* $u = a, b$, *assume* $||\mathbf{Z}^{a,T}\mathbf{Z}^a - \mathbf{Z}^{b,T}\mathbf{Z}^b|| \leq (1 - \gamma)\epsilon$, *then we have:*

$$||\mathbf{Z}^{b,T}(\mathbf{Z}^b\mathbf{Z}^{b,T})^{-1}\mathbf{Z}^b\mathbf{Z}^{a,T} - \mathbf{Z}^{a,T}|| \leq \lambda^{1.5}(\lambda^2 + 1)(1 - \gamma)\epsilon \tag{13}$$

*Proof.* For conciseness, we temporarily denote $\mathbf{A} = \mathbf{Z}^a$ and $\mathbf{B} = \mathbf{Z}^b$. To begin with, we notice:

$$(\mathbf{B}^T(\mathbf{B}\mathbf{B}^T)^{-1}\mathbf{B}\mathbf{A}^T - \mathbf{A}^T)\mathbf{A}\mathbf{A}^T = (\mathbf{B}^T(\mathbf{B}\mathbf{B}^T)^{-1}\mathbf{B} - \mathbf{I})(\mathbf{A}^T\mathbf{A} - \mathbf{B}^T\mathbf{B})\mathbf{A}^T \tag{14}$$

Then, we have:

$$\begin{aligned}
||\mathbf{B}^T(\mathbf{B}\mathbf{B}^T)^{-1}\mathbf{B}\mathbf{A}^T - \mathbf{A}^T|| &\leq ||(\mathbf{A}\mathbf{A}^T)^{-1}|| \cdot ||\mathbf{B}^T(\mathbf{B}\mathbf{B}^T)^{-1}\mathbf{B} - \mathbf{I}|| \cdot ||\mathbf{A}^T\mathbf{A} - \mathbf{B}^T\mathbf{B}|| \cdot ||\mathbf{A}^T|| \\
&\leq \lambda \cdot \left[||\mathbf{B}^T(\mathbf{B}\mathbf{B}^T)^{-1}\mathbf{B}|| + 1\right] \cdot (1 - \gamma)\epsilon \cdot ||\mathbf{A}^T|| \\
&\leq \lambda \cdot \left[||\mathbf{B}\mathbf{B}^T|| \cdot ||(\mathbf{B}\mathbf{B}^T)^{-1}|| + 1\right] \cdot (1 - \gamma)\epsilon \cdot \sqrt{||\mathbf{A}\mathbf{A}^T||} \\
&= \lambda^{1.5}(\lambda^2 + 1)(1 - \gamma)\epsilon
\end{aligned} \tag{15}$$

$\square$

Now, we are ready to prove Theorem 1. For reader's convenience, we re-state Theorem 1 in below.

**Theorem A.1.** *(Crossmodal KD in linear binary classification). Without loss of generality, we assume* $\mathbf{f}_{\boldsymbol{\theta}_t}(\cdot) : \mathcal{X}^a \mapsto \mathcal{Y}$ *and* $\mathbf{f}_{\boldsymbol{\theta}_s}(\cdot) : \mathcal{X}^b \mapsto \mathcal{Y}$. *Suppose* $\max\{||\mathbf{Z}^u\mathbf{Z}^{u,T}||, ||(\mathbf{Z}^u\mathbf{Z}^{u,T})^{-1}||\} \leq \lambda$ *always holds for both* $u = a$ *or* $b$, *and* $\mathbf{g}^u(\cdot)$ *are identity functions. If there exists* $(\epsilon, \delta)$ *such that*

$$Pr\left[||\mathbf{Z}^{a,T}\mathbf{Z}^a - \mathbf{Z}^{b,T}\mathbf{Z}^b|| \leq (1 - \gamma)\epsilon\right] \geq 1 - \delta \tag{16}$$

*Then, with an initialization at* $t = 0$ *satisfying* $R_n^{dis}(\boldsymbol{\theta}_s(0)) \leq q$, *we have, at least probability* $1 - \delta$:

$$R_n^{dis}(\boldsymbol{\theta}_s(t = +\infty)) \leq n\left(\frac{\epsilon^\star}{1 - e^{-\epsilon^\star}} - 1 - \ln\frac{\epsilon^\star}{1 - e^{-\epsilon^\star}}\right) \tag{17}$$

*where* $\epsilon^\star = \lambda^{1.5}(\lambda^2 + 1)(1 - \gamma)\epsilon$ *and* $R_n^{dis}(\boldsymbol{\theta}_s)$ *is the empirical risk defined by KL divergence:*

$$R_n^{dis}(\boldsymbol{\theta}_s(t)) = \sum_{i=1}^{n} -\sigma(\boldsymbol{\theta}_t^T\mathbf{x}_i^a) \cdot \ln\frac{\sigma(\boldsymbol{\theta}_s^T\mathbf{x}_i^b)}{\sigma(\boldsymbol{\theta}_t^T\mathbf{x}_i^a)} - \left[1 - \sigma(\boldsymbol{\theta}_t^T\mathbf{x}_i^a)\right] \cdot \ln\frac{1 - \sigma(\boldsymbol{\theta}_s^T\mathbf{x}_i^b)}{1 - \sigma(\boldsymbol{\theta}_t^T\mathbf{x}_i^a)} \tag{18}$$

*Proof.* First, under the given conditions, if we could prove:

$$||\mathbf{Z}^{a,T}\mathbf{Z}^a - \mathbf{Z}^{b,T}\mathbf{Z}^b|| \leq (1 - \gamma)\epsilon \implies R_n^{\text{dis}}(\boldsymbol{\theta}_s(t = +\infty)) \leq n\left(\frac{\epsilon^\star}{1 - e^{-\epsilon^\star}} - 1 - \ln\frac{\epsilon^\star}{1 - e^{-\epsilon^\star}}\right) \tag{19}$$

Then, adding the outer probability bracket doesn't alter the conclusion, and the theorem is proved. Therefore, in the following, we will focus on proving Eq. (19). The proof consists of two parts. In the first part, we will show that there exists an $\boldsymbol{\theta}_s^\star$, such that $R_n^{\text{dis}}(\boldsymbol{\theta}_s^\star)$ is bounded. Then, in the second part, we will prove that if the training process is long enough (i.e., $t \to +\infty$), then the final distillation risk $R_n^{\text{dis}}(\boldsymbol{\theta}_s(t = +\infty))$ is further bounded by $R_n^{\text{dis}}(\boldsymbol{\theta}_s^\star)$.

Without loss of generality, we assume the trained teacher weight $||\boldsymbol{\theta}_t|| = 1$ since in linear binary classification, scaling the weight doesn't affect the final prediction. Now, let us consider:

$$\boldsymbol{\theta}_s^\star = (\mathbf{Z}^b\mathbf{Z}^{b,T})^{-1}\mathbf{Z}^b\mathbf{Z}^{a,T}\boldsymbol{\theta}_t \tag{20}$$

Then we have:

$$\begin{aligned}
||\mathbf{Z}^{b,T}\boldsymbol{\theta}_s^\star - \mathbf{Z}^{a,T}\boldsymbol{\theta}_t|| &= ||\mathbf{Z}^{b,T}(\mathbf{Z}^b\mathbf{Z}^{b,T})^{-1}\mathbf{Z}^b\mathbf{Z}^{a,t}\boldsymbol{\theta}_t - \mathbf{Z}^{a,T}\boldsymbol{\theta}_t|| \\
&\leq ||\mathbf{Z}^{b,T}(\mathbf{Z}^b\mathbf{Z}^{b,T})^{-1}\mathbf{Z}^b\mathbf{Z}^{a,T} - \mathbf{Z}^{a,T}|| \cdot ||\boldsymbol{\theta}_t|| \\
&\leq \lambda^{1.5}(\lambda^2 + 1)(1 - \gamma)\epsilon
\end{aligned} \tag{21}$$

where in the last line, we have used Lemma 2. Then using Corollary 1, we obtain

$$R_n^{\text{dis}}(\boldsymbol{\theta}_s^\star) = L_n(\mathbf{Z}^{b,T}\boldsymbol{\theta}_s^\star, \mathbf{Z}^{a,T}\boldsymbol{\theta}_t^\star) \le n(\frac{\epsilon^\star}{1 - e^{-\epsilon^\star}} - 1 - \ln\frac{\epsilon^\star}{1 - e^{-\epsilon^\star}}) \tag{22}$$

where $\epsilon^\star = \lambda^{1.5}(\lambda^2 + 1)(1 - \gamma)\epsilon$. Now, the remaining effort is to prove that if time is sufficiently long (i.e., $t \to +\infty$), the trained loss $R_n^{\text{dis}}(\boldsymbol{\theta}_s(t))$ will be smaller than $R_n^{\text{dis}}(\boldsymbol{\theta}_s^\star)$.

To begin with, we apply Theorem A.2 and Corollary A.1 from (Phuong & Lampert, 2019): For any sublevel set $\Theta = \{\boldsymbol{\theta}_s : R_n^{\text{dis}}(\boldsymbol{\theta}_s) \le q\}$, there exists $c > 0$ such that:

$$cR_n^{\text{dis}}(\boldsymbol{\theta}_s) - cR_n^{\text{dis}}(\boldsymbol{\theta}_s^\star) \le \frac{1}{2}||\nabla R_n^{\text{dis}}(\boldsymbol{\theta}_s)||^2 \tag{23}$$

Compared to the original Corollary A.1 of (Phuong & Lampert, 2019), a slight modification is done for it to suit our case. Proving the existence of $c$ is obvious by noticing that the left-hand side of Eq. (50) of (Phuong & Lampert, 2019) is relaxed to $R_n^{\text{dis}}(\boldsymbol{\theta}_s^\star)$ instead of 0 in our case. Next, noticing Eq. (8) and (53) of (Phuong & Lampert, 2019), we have:

$$(R_n^{\text{dis}})' = \frac{dR_n^{\text{dis}}(\boldsymbol{\theta}_s(t))}{dt} = -||\nabla R_n^{\text{dis}}(\boldsymbol{\theta}_s)||^2 \le -2cR_n^{\text{dis}}(\boldsymbol{\theta}_s) + 2cR_n^{\text{dis}}(\boldsymbol{\theta}_s^\star) \tag{24}$$

which could be simplified as (Phuong & Lampert, 2019):

$$\begin{aligned}
\left[R_n^{\text{dis}} - R_n^{\text{dis}}(\boldsymbol{\theta}_s^\star)\right]' &\le -2c\left[R_n^{\text{dis}} - R_n^{\text{dis}}(\boldsymbol{\theta}_s^\star)\right] \\
=> \left\{\ln\left[R_n^{\text{dis}} - R_n^{\text{dis}}(\boldsymbol{\theta}_s^\star)\right]\right\}' &\le -2c
\end{aligned} \tag{25}$$

Integrating over $[0, \tau]$ yields:

$$R_n^{\text{dis}}(\boldsymbol{\theta}_s(\tau)) \le R_n^{\text{dis}}(\boldsymbol{\theta}_s(0)) \cdot e^{-2c\tau} + R_n^{\text{dis}}(\boldsymbol{\theta}_s^\star) \tag{26}$$

Thus, as $\tau \to +\infty$, $R_n^{\text{dis}}(\boldsymbol{\theta}_s(\tau = +\infty))$ will be upper-bounded by $R_n^{\text{dis}}(\boldsymbol{\theta}_s^\star)$. The above equation also indicates the convergence speed. □

**Remarks.** Firstly, the above theorem implies that the training distillation loss is upper-bounded by a monotonic function with respect to $\gamma$. When $\gamma$ increases from 0 to 1, the upper bound gradually decreases. Further combining the above theorem with Rademacher complexity, we could even obtain a bound on the generalization error. Nevertheless, the key component in the generalization bound is already shown here. As such, readers should be alerted that our derived theorem is primitive, that our main contribution in this paper are MVD and MFH, and that a complete theoretical analysis itself could be a standalone work. Secondly, we emphasize that in proving our theorem, the MVD decision rule shown in Eq. (3) is not utilized. In the context of our theorem, Eq. (3) states that there exist $\boldsymbol{\theta}_s$ and $\boldsymbol{\theta}_t$, which could make the corresponding student and teacher network achieve zero generalization error [4]. However, our focus in the proved theorem is merely the training distillation loss, neither involving the generalization error, nor the cross entropy loss between the student prediction and the true label, so this MVD decision rule is not used. However, the decision rule is essential to fully characterize the MVD model as it states the *modality-general* and *modalty-specfic decisive* features are sufficient to determine the label, and we expect it will be of use when proving the generalization error. Finally, with the above two remarks on our insufficiency, it would be of great interest to extend our theorem to the generalization error with a linear (or even non-linear) $\mathbf{g}^u(\cdot)$ and $\rho \ne 0$ in $K$-class classification. In a nutshell, we hope our proposed MVD could work as a powerful model being utilized to theoretically prove generalization error of crossmodal KD.

## B  EXPERIMENTAL SETUP AND MORE RESULTS

### B.1  CASE STUDIES IN SEC. 3

In this section, we describe implementation details and provide more results for the two case studies in Sec. 3. AV-MNIST (Vielzeuf et al., 2018) is an audio-visual dataset created by pairing audio and image features. The two modalities are MNIST images with 75% energy removed by principal

---

[4]However, such a $\boldsymbol{\theta}_s$ might not able to achieve when doing crossmodal KD by using Eq. (1.

component analysis and audio spectrograms with random natural noise injected. There are 50,000 pairs for training, 5,000 pairs for validation and 10,000 pairs for testing. Following (Vielzeuf et al., 2018; Gao et al., 2022), we adopt a 6-layer CNN as the audio teacher network. The audio student network is implemented as a 3-layer CNN, and the multimodal teacher is a late fusion network. The multimodal teacher uses LeNet5 (LeCun et al., 1989) as the image backbone and a 5-layer CNN as the audio backbone; Audio and image features are then concatenated and passed to fully-connected layers for the final prediction. Recall that $\rho$ in Eq. (1) in the main text controls the relative importance of the two loss terms when training the student network. We experiment with both $\rho = 0$ and $\rho = 0.5$, and repeat the experiments for 10 times. We have provided the results of $\rho = 0.5$ in Table 1 in the main text, and a more detailed version can be found in Table 5 below.

Table 5: Evaluation of unimodal KD (UM-KD) and crossmodal KD (CM-KD) on AV-MNIST.

|  | Teacher | | Student | | |
|  | Mod. | Acc. (%) | Mod. | Acc. (%) | |
|  |  |  |  | $\rho = 0$ | $\rho = 0.5$ |
| No-KD | - | - | A | $68.36 \pm 0.79$ | $68.36 \pm 0.79$ |
| UM-KD | A | 84.57 | A | $69.86 \pm 0.70$ | $70.10 \pm 0.50$ |
| CM-KD | I + A | 91.61 | A | $69.46 \pm 1.12$ | $69.73 \pm 0.73$ |

From the table, we can see that crossmodal KD does not have advantages over unimodal KD for both values of $\rho$. We hypothesize that the proportion of modality-general decisive information is small in this dataset since a multimodal data pair is assembled by randomly pairing an image with an audio that belongs to the same class. Thus the two modalities are not naturally correlated and there may be little modality-general information. MFH provides a plausible explanation for this failure case of crossmodal KD.

NYU Depth V2 (Nathan Silberman & Fergus, 2012) contains 1,449 aligned RGB and depth images with 40-class labels, where 795 images are used for training and 654 images are for testing. $\rho$ is set to 0.5. We implement two model architectures for the multimodal teacher: (1) Channel Exchanging Networks (CEN) (Wang et al., 2020) and (2) Separation-and-Aggregation Gate (SA-Gate) (Chen et al., 2020b). The unimodal teacher and student are adopted as the RGB branch of the corresponding multimodal network. The results are shown in Table 6, part of which corresponds to the right section of Table 1 in the main text.

Table 6: Evaluation of unimodal KD (UM-KD) and crossmodal KD (CM-KD) on NYU Depth V2.

|  | CEN | | | | SA-Gate | | | |
|  | Teacher | | Student | | Teacher | | Student | |
|  | Mod. | mIoU | Mod. | mIoU | Mod. | mIoU | Mod. | mIoU |
| No-KD | - | - | RGB | 45.69 | - | - | RGB | 46.36 |
| UM-KD | RGB | 45.69 | RGB | 46.23 | RGB | 46.36 | RGB | 48.00 |
| CM-KD | RGB + D | 51.14 | RGB | 46.70 | RGB + D | 51.00 | RGB | 47.78 |

Table 6 demonstrates that crossmodal KD is not effective in both cases. The great advantages in teacher performance does not enhance student performance. Adopting CEN as the multimodal teacher seems better than SA-Gate, but the improvement compared with unimodal KD is still marginal (*i.e.*, from 46.23% to 46.70%). According to MFH, different teacher networks utilize different amount of *modality-general decisive features* for prediction, which results in different distillation performance. We hypothesize that CEN has a larger $\gamma$ than SA-Gate due to their model design: CEN shares all parameters for the RGB and depth input except for Batch Normalization layer while SA-Gate has separate encoders for the two modalities. This indicates that CEN is more modality-general than SA-Gate, and this may further account for their performance differences. There may be other factors lying behind, and one future direction is to develop methods to compare existing model architectures to find a teacher architecture that best suits crossmodal KD.

## B.2 SYNTHETIC GAUSSIAN

Assume two vectors $\mathbf{x}^a \in \mathbb{R}^{d_1}$ and $\mathbf{x}^b \in \mathbb{R}^{d_2}$ compose one multimodal data pair $(\mathbf{x}^a, \mathbf{x}^b)$. We select a subset of input features as *decisive features*, denoted by $\mathbf{x}^* \in \mathbb{R}^d$. We assume that $\mathbf{x}^*$ exist in both $\mathbf{x}^a$ and $\mathbf{x}^b$, and denote the corresponding decisive feature index set of $\mathbf{x}^a$ ($\mathbf{x}^b$) as $J_1$ ($J_2$). The separating hyperplanes are denoted by $\boldsymbol{\delta} \in \mathbb{R}^d$. Formally, we generate one feature-label pair $(\mathbf{x}^a, \mathbf{x}^b, y)$ by:

$$
\begin{aligned}
\mathbf{x}^* &\sim \mathcal{N}(\mathbf{0}, \mathbf{I}_d), & y &\leftarrow \mathbb{1}(\langle \boldsymbol{\delta}, \mathbf{x}^* \rangle > 0) \\
\mathbf{x}^a &\sim \mathcal{N}(\mathbf{0}, \mathbf{I}_{d_1}), & \mathbf{x}^a_{J_1} &\leftarrow \mathbf{x}^*_{J_1} \\
\mathbf{x}^b &\sim \mathcal{N}(\mathbf{0}, \mathbf{I}_{d_2}), & \mathbf{x}^b_{J_2} &\leftarrow \mathbf{x}^*_{J_2}
\end{aligned}
\tag{27}
$$

As depicted in MVD, *modality-general decisive features* are *decisive features* shared by two modalities and thus indexed by $J_1 \cap J_2$. $J_1 \cup J_2$ represents the index set of *decisive features* from both modalities. Therefore, $\alpha = 1 - \frac{|J_2|}{|J_1 \cup J_2|}$, $\beta = 1 - \frac{|J_1|}{|J_1 \cup J_2|}$ and $\gamma = \frac{|J_1 \cap J_2|}{|J_1 \cup J_2|}$. By changing $J_1$ and $J_2$, we can generate multimodal data with different inherent characteristics (*i.e.*, different $\alpha$, $\beta$, and $\gamma$). We consider two settings: (1) varying $\gamma$ (Fig. 2 in the main paper). Let $d_1 = 25$, $d_2 = 50$ and $d = 20$, we gradually increase $|J_1 \cap J_2|$ from 0 to 10, with a step size of 2 and perform KD on every step. Consequently, $\gamma$ takes the value of [0, 0.11, 0.25, 0.43, 0.67, 1], and $\alpha = \beta = \frac{1-\gamma}{2}$. (2) varying $\alpha$ (Fig. 3 in the main text). Let $d_1 = d_2 = 50$ and $d = |J_1 \cup J_2|$ increase from 10 to 50, with a step size of 10. Thus, $\alpha$ takes the value of [0, 0.5, 0.67, 0.75, 0.80, 0.83]. We set $\beta$ to be 0 through the process, and $\gamma = 1 - \alpha$.

Following (Lopez-Paz et al., 2015), the teacher and the student are both implemented as logistic regression models, and we use 200 samples for training and 1,000 samples for testing. $\boldsymbol{\delta}$ is sampled from the standard normal distribution. $\rho$ in Eq. 1 in the main text is set as 0.5. Results are averaged over 10 runs.

## B.3 MM-IMDB

MM-IMDB (Arevalo et al., 2017) is the largest publicly available multimodal dataset for genre prediction on movies. It contains 25,959 movie titles and posters that belong to 27 movie genres. We pick two movie genres (*i.e.*, drama and comedy) for multi-label classification. There are 15,552 data for training, 2,608 for validation, and 7,799 for testing. We adopt the same pre-processing method as in (Arevalo et al., 2017) to extract image and text features. We consider the special case of crossmodal KD, where the teacher is a multimodal network that takes both images and text as input and student is a unimodal text network. The unimodal and multimodal architecture are identical to the one in (Liang et al., 2021). As described in Sec. 5.1, we experiment with two teacher networks that have the same architecture but differ in $\gamma$: (1) We regularly train a multimodal network with labels; (2) Following (Xue et al., 2021), we train a multimodal network that receives pseudo labels from an unimodal image network. The second teacher only has access to pseudo labels from the image modality and leans towards the image modality when giving predictions. In other words, it is more modality-specific (*i.e.*, has a smaller $\gamma$) than the regular teacher. We randomly split training data with the ratio 50%:50%, and use the first half to train the unimodal teacher and the general multimodal teacher. The other part of data is used for training the student network, and we set $\rho = 0$.

Table 7: Results on MM-IMDB movie genre classification. T and I represent text and images, respectively. With identical architecture and similar accuracy, a modality-specific teacher leads to worse crossmodal KD performance than a regular teacher since it utilizes fewer *modality-general decisive features* for prediction.

| | Teacher | | | Student | | |
|---|---|---|---|---|---|---|
| | Mod. | micro F1 (%) | macro F1 (%) | Mod. | micro F1 (%) | macro F1 (%) |
| UM-KD | T | 61.76 | 61.13 | T | 61.04 | 56.44 |
| Modality-specific-KD | T+I | 62.01 | 61.08 | T | 61.77 | 57.26 |
| Regular-KD | T+I | 61.01 | 60.37 | T | 65.09 | 63.31 (↑) |

Table 7 shows the teacher and student performance for both unimodal KD and crossmodal KD. We select three teacher models that have similar performance on test data, and use them for distillation to detach the influence of teacher performance. Clearly, the three teachers transfer different knowledge to the student. The unimodal teacher comes from the image modality and the modality-specific multimodal teacher is also biased towards the image modality due to its training strategy. Finally, the regular multimodal teacher adopts more modality-general information compared with the previous two teachers (*i.e.*, has a larger $\gamma$). As can be seen from the table, it results in the best unimodal text student, which helps verify our proposed MFH.

### B.4  RAVDESS AND VGGSOUND

We first present a permutation-based approach to rank given multimodal features according to the amount of modality-general decisive information available in each feature channel. This approach offers an alternative way to obtain teachers of different $\gamma$ and helps validate the proposed MFH. As described in Sec. 5.4, once we have a sorted list for all feature channels, we can derive a modality-general (or modality-specific) teacher by nullifying the top $r\%$ smallest (or largest) channels during the distillation process.

The major steps of our proposed feature ranking approaches are demonstrated in Algorithm 1. The input of Algorithm 1 are $\mathbf{X}^a \in \mathbb{R}^{n \times d_1}$, $\mathbf{X}^b \in \mathbb{R}^{n \times d_2}$, and $\mathbf{Y} \in \mathbb{R}^n$, representing $n$ paired features from modality $a$ and $b$, and $n$ target labels, respectively. The output is a salience vector $\mathbf{p} \in \mathbb{R}^{d_1}$ for *modality-specific decisive* features in modality $a$, where its $i$-th entry $p_i \in [0, 1]$ reflects the salience of the $i$-th feature dimension. A larger salience value indicates a more *modality-general decisive* feature channel.

---

**Algorithm 1** Modality-General Decisive Feature Ranking

---

**Input:** multimodal features ($\mathbf{X}^a \in \mathbb{R}^{n \times d_1}, \mathbf{X}^b \in \mathbb{R}^{n \times d_2}, \mathbf{Y} \in \mathbb{R}^n$)
**Output:** salience vector $\mathbf{p} \in \mathbb{R}^{d_1}$ for features of modality $a$
 1: Jointly train two unimodal networks $\mathbf{f}_{\boldsymbol{\theta}_1^*}$ and $\mathbf{f}_{\boldsymbol{\theta}_2^*}$ using the following loss:

$$\min_{\boldsymbol{\theta}_1, \boldsymbol{\theta}_2} \mathcal{L} = Dist(\mathbf{f}_{\boldsymbol{\theta}_1}(\mathbf{X}^a), \mathbf{f}_{\boldsymbol{\theta}_2}(\mathbf{X}^b)) + CE(\mathbf{Y}, \mathbf{f}_{\boldsymbol{\theta}_1}(\mathbf{X}^a)) + CE(\mathbf{Y}, \mathbf{f}_{\boldsymbol{\theta}_2}(\mathbf{X}^b)) \qquad (28)$$

$\qquad \triangleright Dist(\cdot, \cdot)$ denotes a distance loss (*e.g.*, mean squared error) and $CE$ denotes cross entropy
 2: **for** $i = 1$ to $d_1$ **do**  $\qquad \qquad \qquad \triangleright$ Calculate the salience for the $d$-th feature dimension
 3: $\quad$ $p_i = 0$
 4: $\quad$ **for** $m = 1$ to $M$ **do**  $\qquad \qquad \qquad \triangleright$ Repeat permutation $M$ times for better stability
 5: $\qquad$ permute the $i$-th column of $\mathbf{X}^a$ yielding $\tilde{\mathbf{X}}^a$
 6: $\qquad$ $p_i = p_i + \frac{1}{M} \times Dist(\mathbf{f}_{\boldsymbol{\theta}_1^*}(\tilde{\mathbf{X}}^a), \mathbf{f}_{\boldsymbol{\theta}_2^*}(\mathbf{X}^b))$
 7: $\quad$ **end for**
 8: **end for**
 9: Perform normalization: $\mathbf{p} = \frac{\mathbf{p}}{\max_i p_i} \in [0, 1]^{d_1}$

---

Since input-level features contain much label-irrelevant noise, our Algorithm 1 is designed following a trace-back thought starting from the output level. Namely, we drive two unimodal networks to the state of "feature alignment" at the output level using Eq. (28), and then use permutation to identify which input feature dimension has a larger impact to the state. Those more influential to the state (*i.e.*, a large distance in step 6) will be assigned a larger salience value.

Table 8: Test Accuracy (%) on RAVDESS emotion recognition. Teacher modality is audio and student modality is images. Modality-general crossmodal KD demonstrates best performance for all feature nullifying dimension ratio $r\%$.

| $r$ (%) | Regular | Modality-specific | Random | Modality-general |
|---|---|---|---|---|
| 25 | 77.22 | 75.26 ($\downarrow$) | 77.24 | 78.12 ($\uparrow$) |
| 50 | 77.22 | 42.66 ($\downarrow$) | 64.75 | 78.28 ($\uparrow$) |
| 75 | 77.22 | 12.00 ($\downarrow$) | 63.40 | 77.82 ($\uparrow$) |

In step 1, we jointly train two unimodal networks $\mathbf{f}_{\boldsymbol{\theta}_1^*}$ and $\mathbf{f}_{\boldsymbol{\theta}_2^*}$ that respectively take unimodal data $\mathbf{X}^a$ and $\mathbf{X}^b$ as input. The first loss term in Eq. (28) aims to align feature spaces learned by the two networks, and the remaining loss terms ensure that learned features are essential for a correct prediction. We believe that this training strategy aligns three sources of *decisive features* at the output level. In step 2, we follow the idea of permutation feature importance (Breiman, 2001) to trace back *modality-general decisive features* at the input level. For the $i$-th dimension in $\mathbf{X}^a$, we randomly permute $\mathbf{X}^a$ along this dimension and obtain a permuted $\tilde{\mathbf{X}}^a$ in step 5. Next, we calculate the distance between $\mathbf{f}_{\boldsymbol{\theta}_1^*}(\tilde{\mathbf{X}}^a)$ and $\mathbf{f}_{\boldsymbol{\theta}_2^*}(\mathbf{X}^b)$ in step 6. A large distance indicates that the $i$-th dimension largely influences the state of "feature alignment". Consequently, we are able to quantify the proportion of *modality-general decisive features* in each input feature channel and use the salience vector $\mathbf{p}$ to represent it. We repeat the permutation process for $M$ times and average the distance value for good stability. Finally, $\mathbf{p}$ is normalized to $[0, 1]^{d_1}$ in step 9.

Note that: (1) The focus of this paper is to propose and validate MFH. Algorithm 1 presents an approach to rank features and allows us to derive teachers of different $\gamma$ to justify the MFH implication. Developing methods that can separate *modality-general decisive features* and *modality-specific decisive features* is a challenging problem worth deep investigation and is left as future work. (2) Algorithm 1 is not limited to feature ranking at the input level. $\mathbf{X}^a$ and $\mathbf{X}^b$ can be features extracted from middle layers of the neural network as well. In such case, output $\mathbf{p}$ reflects the salience for each middle-layer feature channel. (3) Algorithm 1 could be equally applied to rank *modality-general decisive features* for modality $b$ as long as we permute $\mathbf{X}^b$.

Next, we present results by applying this feature ranking approach to two multimodal applications, *i.e.*, RAVDESS emotion recognition and VGGSound event classification.

The Ryerson Audio-Visual Database of Emotional Speech and Song (RAVDESS) (Livingstone & Russo, 2018) contains videos and audios of 24 professional actors vocalizing two lexically-matched statements. For modality $a$ (*i.e.*, teacher modality), we adopt Kaiser best sampling and take mel-frequency cepstral coefficients (MFCCs) features from corresponding audio. For modality $b$ (*i.e.*, student modality), we uniformly sample single-frame images every 0.5 second from each video. We randomly split image-audio pairs, and have 7,943 data for training, 2,364 data for validation and 1,001 data for testing. Similar to (Xue et al., 2021), the teacher and student architecture are 3-layer CNNs followed by 3 fully-connected layers. We set $\rho$ in Eq. (1) in the main text as 0 (*i.e.*, only use $\mathcal{L}_{kd}$ for distillation) to fully observe the teacher's influence on student performance. We report results with three feature nullifying ratio $r\%$ in Table 8, which is a detailed version of Table 4 in the main text. Results are averaged over 5 runs.

As shown in the table, with more feature channels getting nullified (*i.e.*, increasing $r\%$), the random and modality-specific version both suffer from a heavy performance degradation. On the contrary, a modality-general teacher still attains satisfactory distillation performance and outperforms regular KD even when the feature nullifying ratio goes to 75%. This demonstrates the efficacy of our proposed feature ranking method as well as the practical value of MFH.

Table 9: mean Average Precision (%) on VG-GSound event classification. Teacher modality is audio and student modality is video.

| $\rho$ | 0 | 0.5 |
|---|---|---|
| Regular | 30.62 | 30.70 |
| Modality-specific | 24.98 ($\downarrow$) | 28.14 ($\downarrow$) |
| Random | 28.99 | 29.56 |
| Modality-general | 31.88 ($\uparrow$) | 31.98 ($\uparrow$) |

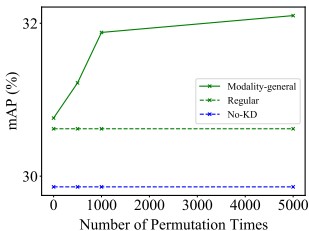

Figure 6: Crossmodal KD peprformance with varying permutation number $M$.

VGGSound is a large-scale audio-visual correspondent dataset. We randomly choose 100 class from its 310 classes and obtain 56,614 audio-video pairs for training and 4,501 audio-video pairs for testing. Videos clips and audio spectrograms are taken as input features, respectively. The audio network is implemented as a ResNet-18 / ResNet-50 backbone followed by linear layers and the video network is the same architecture with 2D convolution replaced by 3D convolution. For Table 4 in the main paper, we set $\rho$ in Eq. (1) to be 0 and experiment with both ResNet-18 and ResNet-50. In

Table 9, we report results of both $\rho = 0$ and $\rho = 0.5$ with the ResNet-18 backbone. The conclusion is consistent: a modality-general teacher improves student performance while a modality-specific teacher results in performance degradation. These results help validate our proposed MFH.

In Algorithm 1, we repeat permutation $M$ times for a better estimation of each feature dimension's salience value. Fig. 6 provides an analysis on the number of permutation times $M$. As $M$ increases, we have a more accurate estimation of $\mathbf{p}$, so modality-general KD gradually improves and finally reaches a plateau.

## C    COMPARISON OF MVD WITH THE MULTI-VIEW ASSUMPTION

Below we compare MVD with the multi-view assumption (Sridharan & Kakade, 2008). We adopt the same notations used in the main text here for the multi-view learning paradigm: assume the input variable is partitioned into two different views $\mathbf{x}^a$ and $\mathbf{x}^b$, and there is a target variable $y$ of interest. The multi-view assumption states that either view alone is sufficient to predict the target label $y$ accurately. As illustrated in Figure 7 (a), all task-relevant information is assumed to lie in the shared regions between $\mathbf{x}^a$ and $\mathbf{x}^b$.

Despite its wide use in unimodal self-supervised learning, the multi-view assumption is likely to fail when $\mathbf{x}^a$ and $\mathbf{x}^b$ are from different modalities (Tsai et al., 2020). In fact, whether the multi-view assumption holds is largely dependent on downstream tasks and modalities involved. On one hand, for tasks such as image classification, either the image itself or its caption is sufficient to derive the target label. On the other hand, for tasks such as MM-IMDB movie genre classification, using the movie poster (image modality) alone may not convey which category this movie belongs to. Movie descriptions (text modality-specific information) are also needed to infer the target label $y$. Similarly, using the depth image alone is not sufficient for RGB-D semantic segmentation as there are regions in the image that have the same depth values but correspond to different semantic labels. The multi-view assumption thus does not hold for many challenging multimodal tasks.

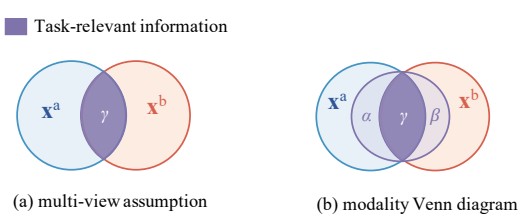

(a) multi-view assumption

(b) modality Venn diagram

Figure 7: Illustration of (a) the multi-view assumption and (b) our modality Venn diagram.

Our proposed MVD generalizes the multi-view assumption by taking modality-specific decisive features into consideration as well. It states that task-relevant information is composed of three parts: (1) modality-general decisive features; (2) decisive features specific to modality $a$; and (3) decisive features specific to modality $b$, as illustrated in Figure 7 (b). Consequently, the multi-view assumption can be considered as a special case of MVD when $\alpha = \beta = 0$ and $\gamma = 1$.

