# OpenReview forum: "The Modality Focusing Hypothesis: Towards Understanding Crossmodal Knowledge Distillation"
_ICLR.cc/2023/Conference — ICLR 2023 notable top 5%_

### Official Review · Reviewer_i93D · 2022-10-15

**Confidence:** 4
**Clarity, Quality, Novelty And Reproducibility:** Good quality and clarity.
**Correctness:** 4
**Technical Novelty And Significance:** 4
**Empirical Novelty And Significance:** 3
**Recommendation:** 8

**Strength And Weaknesses:**

Pros:
1. The proposed Modality Focusing Hypothesis is significant and might raise some interest in controllable cross-model distillation. Rich experiments verify this claim.
2. The organization of the paper is good in general.

Cons:
1. Although I admit and like the Modality Focusing Hypothesis, this hypothesis conflicts with the general multi-view assumption, i.e., each view could bring extra useful downstream information [1]. Different views always have some specific information and we always find that learning a multi-modal model is better than every single modality (CLIP), at least in most datasets. Why this phenomenon is different in knowledge distillation? One might argue that view-specific information contains much redundancy [1]. I'd like to see some explanations about this point.
2. It is not clear to me that how to control the $\gamma$, e.g., how to calculate $\gamma$ in Table 2? What's the rule? Also, how to quantize this parameter during other experiments?
3. For MM-IMDB data, why there is more modality-specific information?
4. The modality Venn diagram is not new enough, as there are plenty of works that use this to analyze the effectiveness of multi-modal learning [1-3]. Specifically, [2] also claims that modality-general information is essential. The author is encouraged to change their claim that treats this as a contribution and do a better job by clarifying the differences between these relevant works and the proposed method.

[1] Self-supervised Learning from a Multi-view Perspective, ICLR'21

[2] Dual Contrastive Prediction for Incomplete Multi-View Representation Learning, TPAMI'22

[3] COMPLETER: Incomplete Multi-view Clustering via Contrastive Prediction, CVPR'21

**Summary Of The Paper:**

This paper gives some new insight into the multi-modal knowledge distillation community. Specifically, teacher accuracy does not necessarily indicate student performance, while modality-general features are the key in crossmodal KD. With the explanation of the modality Venn diagram, such a claim is straightforward and easy to understand. The authors also do some controlled experiments and try to support their analysis.


**Summary Of The Review:**

I recommend accepting the paper (rating 6). I liked the hypothesis proposed by the authors. Although this paper has some drawbacks, the contribution is enough. Ratings can be improved further if the authors could solve my questions.

----------- UPDATE I -----------

I thank the authors for their response to my concerns, especially about the **modality Venn diagram** and more experimental details. After reading the response, I consistently consider the proposed hypothesis is well-motivated.

I hope the authors could highlight the details about the $\alpha,\beta,$ and $\gamma$ in each experiment (e.g., bold, color) for better illustration.

---

> ### Author Response · Authors · 2022-11-16
> **Response to Reviewer i93D (1/2)**
>
> We thank Reviewer i93D for the positive comments and for providing thoughtful feedback on our work. Below please see our detailed responses.
>
> **Comparison of the modality Venn diagram with multi-view assumption**
>
> > this hypothesis conflicts with the general multi-view assumption, i.e., each view could bring extra useful downstream information”
>
> We would like to clarify that our hypothesis does not really conflict with the multi-view assumption. In fact, the multi-view assumption can be seen as a special case of the modality Venn diagram when $\gamma = 1$.
>
> The multi-view assumption [1,4] states that either view alone is sufficient for the downstream tasks. While this is generally true when the two views are input data and its augmented version in the unimodal context (e.g., an image and the same image with augmentation, non-masked words and the masked words), the assumption does not always hold when the two views are from different modalities. As stated in [1]:
>
> > Under this cross-modality setting, …, the task-relevant information may not mostly lie in the shared information between the input and the self-supervised signal, …, the multi-view assumption is likely to fail.”
>
> For multimodal data, whether this assumption holds true (i.e., all task-relevant information lies in the shared region) is largely dependent on the downstream tasks and modalities. For tasks such as image classification, either the image itself or its caption is sufficient to derive the target label; However, if we consider the MM-IMDB movie genre classification task, by looking at the movie poster (i.e., image), we may not have an idea of which category this movie belongs to and need to resort to movie descriptions (text modality-specific information). Similarly, using the depth image alone is not sufficient for RGB-D semantic segmentation as there are regions in the image that have the same depth values but correspond to different semantic labels.
>
> From the examples above, we can find that our modality Venn diagram (MVD) is a generalization of the multi-view assumption: MVD uses $\alpha$ and $\beta$ to characterize the amount of modality-specific decisive information, whereas in the multi-view assumption, $\alpha$ and $\beta$ are assumed to be 0 as it states that all task-relevant information is modality-general (so $\gamma=1$).
>
> > We always find that learning a multi-modal model is better than every single modality (CLIP), at least in most datasets.”
>
> The inherent difference of MVD with the multi-view assumptions provide explanations for their performance differences. Note that in both synthetic and real-world experiments, we do observe better crossmodal-KD performance than the no-KD baseline when $\gamma$ is large (for instance, the rightmost illustration in Fig. 2, when $\gamma=1$, the student benefits from multimodal information and increases its accuracy by 2.1\%); these results align with the performance boost we observe in many multimodal literatures. In addition, we complete the full story by also considering the scenarios when $\gamma$ is small (i.e., many task-relevant information is specific to one modality), and we observe inferior student performance with modality-specific KD across our experiments, which helps validate the proposed hypothesis.
>
> To sum up, as the multi-view assumption (i.e., $\gamma = 1$) does not hold in many multimodal datasets, our modality focusing hypothesis is based on a more general assumption (i.e., MVD). In this paper, we have investigated crossmodal distillation beyond $\gamma = 1$, and find that the performance variations aligns with the value of $\gamma$ across all experiments: large $\gamma$ (modality-general KD) leads to improvement over regular KD and small $\gamma$ (modality-specific KD) results in performance degradation.
>
> > The author is encouraged to change their claim that treats this as a contribution and do a better job by clarifying the differences between these relevant works and the proposed method.”
>
> Finally, we would like to thank Reviewer i93D for pointing out this important line of multi-view learning literatures [1-3] for our reference. We have added Sec 2.3 and Appendix C to discuss these relevant papers and explain our differences with them in the revised paper. Also, we have revised our claim and make it clear that MVD is built upon previous multi-view and multimodal learning works to characterize multimodal data relations (see Section 2.3 of the revised paper).

---

> > ### Author Response · Authors · 2022-11-16
> > **Response to Reviewer i93D (2/2)**
> >
> >
> > **Experimental Details**
> >
> > > It is not clear to me how to control the $\gamma$. e.g., how to calculate $\gamma$ in Table 2? how to quantize this parameter during other experiments?
> >
> > For the synthetic experiments, we describe the data generation rules in Eq. (27), and $\gamma$ can be calculated based on the decisive feature index set for each modality, i.e., $J_1$ and $J_2$ (details in Appendix B.2). In addition, we have attached the code in the supplementary materials, which involves steps on how to vary $\gamma$ and conduct crossmodal knowledge distillation.
> >
> > For real-world experiments, as explained in Section 4.3, the focus of this paper is to validate the proposed hypothesis rather than develop methods to separate modality-general decisive features for real-world multimodal data. Therefore, we utilize the implication derived from the hypothesis as the basis to design experiments. The overall idea is to obtain teachers of different $\gamma$ and observe the corresponding crossmodal KD performance to verify our hypothesis. In terms of how to vary $\gamma$ in the teacher network, we adopt four different methods based on previous works (details in Section 5.1). Despite the variability in approaches and datasets, we find the results consistently aligned with our modality focusing hypothesis. Moreover, if published, we will release the full code for all experiments with detailed steps on how to derive different $\gamma$ for the teacher network.
> >
> > > For MM-IMDB data, why there is more modality-specific information?
> >
> > Following the discussion above, we have adopted different approaches to obtain teacher models of different $\gamma$. For MM-IMDB, we follow the approach in [5] to train a multimodal teacher; due to its training way (details in [5]), the multimodal teacher is biased towards the image modality, and thus has more image modality-specific information than a regular teacher. Therefore we term it as modality-specific-KD in Table 7.
> >
> > We again thank Reviewer i93D for reviewing our manuscript, and we hope that the above responses adequately address all concerns.

---

> ### Author Response · Authors · 2022-11-21
> **Follow-up Response to Reviewer i93D**
>
> Thank you very much for your updated comments!  We are encouraged to see that our responses have addressed your concerns and help increase the rating. We will keep your new comment in mind and highlight experimental details about $\alpha$, $\beta$ and $\gamma$ clearly in the next version.
>
> Thanks again for your review of our paper. Your questions helped improve our paper! Please let us know if you have any more questions and we are very happy to follow up.

---

### Official Review · Reviewer_vsTN · 2022-10-27

**Confidence:** 2
**Correctness:** 3
**Technical Novelty And Significance:** 3
**Empirical Novelty And Significance:** 3
**Recommendation:** 8

**Clarity, Quality, Novelty And Reproducibility:**

The motivation of this paper is clear, and the way for setting up the hypothesis, theoretical and experimental analysis are reasonable, and easy to follow. And the final results also seem to support the hypothesis. The main problem is how to determine the modality-general decisive features in real applications is not explained well. The results were not difficult to reproduce.

**Strength And Weaknesses:**

The main strength: authors proposed the concept of modality-general decisive features for cross modal knowledge distillation, and proposed a hypothesis that the key factor for improving the student model performance is that the teacher model is trained based on the modality-general decisive features. Authors provided theoretical and experimental analysis to support their hypothesis. This hypothesis could provide new insight for the reason of when a student model could perform well in cross-modal knowledge distillation.

Main weakness: the way to decide what kind of features are modality-general decisive features are difficult to control and search. The advantage of deep learning is to automatically learn the "useful" features for improving the performance, while authors strategy of determining the modality-general decisive features is heuristic, and usually difficult to decide. In particular, how to decide the modality-general decisive features itself is a difficult problem.

**Summary Of The Paper:**

In this paper, authors investigated the key factors that could improve the performance of student model from a teacher model in a cross-modal knowledge distillation set. Based on authors investigation authors proposed a hypothesis that high performance of a teacher model does not always bring high performance student model, and the performance of the student model is determined by the modality-general decisive features. Based on the hypothesis, authors defined index of the modality-general decisive feature, and provided theoretical analysis and experimental evidence to prove their hypothesis.

**Summary Of The Review:**

Authors investigated the key factors in determining when the student model could benefit from a teacher model in cross-modal knowledge distillation. Based on authors investigation, authors proposed a hypothesis with a new concept as modality-general decisive features which is essential in training a teacher model. However, it seems that how to determine what are the modality-general decisive features is not a easy task in real applications.

---

> ### Author Response · Authors · 2022-11-16
> **Responses to Reviewer vsTN**
>
> We thank Reviewer vsTN for reviewing our paper and providing helpful comments on our work. Below please see our detailed responses.
>
> **How to determine the modality-general decisive features in real applications**
>
> > while authors strategy of determining the modality-general decisive features is heuristic,..., how to decide the modality-general decisive features itself is a difficult problem.
>
> We agree with Reviewer vsTN that deciding the modality-general decisive features itself is an open research problem. While our methods are heuristic-based, we can indeed find meaningful modality-general decisive features for real-world multimodal data — we conduct experiments on datasets such as RAVDESS, VGGSound, NYU Depth V2 and MM-IMDB, which all consist of multimodal data collected from different *real-world* multimodal applications. The results across datasets consistently show that increasing $\gamma$ (i.e., the relative proportion of modality-general decisive features) can improve crossmodal KD and decreasing $\gamma$ leads to worse crossmodal KD performance. Moreover, we have provided visualization of modality-general decisive features identified by our methods on VGGSound (a large-scale real-world audio-visual dataset) in Figure 4. From the figure, we can see that modality-general decisive features actually correspond to the area of vocalization, which makes sense as they represent common knowledge possessed by both the audio and visual modality. *All these results support our ability to find modality-general decisive features from real-world multimodal data.*
>
> So far, there hasn’t been a unified method in terms of how to separate common information among all modalities. We notice that many relevant works have touched upon the idea of *modality-shared information*. For instance, multi-view learning literatures (pointed out by Reviewer i93D) [2-3] identify its essential role in incomplete multi-view representation learning; a few multimodal works [6-8] aim to decompose modality-invariant and modality-specific representations for better multimodal network performance. TCGM [9] proposes to maximize modality shared information in the context of semi-supervised multimodal learning. *Each of these works proposes their own methods to decide the modality-general features for their analysis.*
>
> Since there is no well-established solution to identify modality-general decisive features, we tried our best to be inclusive in our experiments: we adopted four different methods to decide $\gamma$. The variability in approaches prevents our analysis from being biased towards one specific method and helps demonstrate the validity of our proposed modality focusing hypothesis. Details of the four methods we adopt are described in Secion 5.1 and Appendix B. As reviewed above, the problem of determining modality-general decisive features clearly deserves further research effort, and we thank Reviewer vsTN for providing this valuable comment as our future direction. Also, we have updated the related work section (specifically, second paragraph of Sec 2.3) to provide more background information about *modality-general information* in the revised version.
>
> Finally, we would like to clarify that the focus of this paper is not to develop methods that can separate modality-general decisive features, but to propose the modality focusing hypothesis, which identifies the importance of modality-general decisive features in crossmodal distillation. Our contributions lie in: (1) we present the modality Venn diagram and formally define modality-specific and modality-general decisive features (no work has done it before); (2) we propose the modality focusing hypothesis, which points out that  modality-general decisive features are the key in crossmodal distillation and hence provides a unique perspective to understand crossmodal distillation (this is the first to investigate the efficacy of knowledge distillation in multimodal learning); (3) we provide theoretical proof (Theorem 1) and conduct experiments on 6 multimodal datasets to validate our proposed hypothesis. In all, this work makes one important first step towards understanding crossmodal distillation, and we hope that it can facilitate the understanding of KD and inspire follow-up ideas (such as finding better ways to determine modality-general decisive features).
>
> We again thank Reviewer vsTN for reviewing our manuscript, and we hope that the above responses adequately address all concerns.

---

> > ### Comment · Reviewer_vsTN · 2022-12-08
> > **Thanks for authors' reply.**
> >
> > Based on authors' reply, I could understand the main focus of the study. Based on this consideration, I will agree to the acceptance (rate 6)

---

### Official Review · Reviewer_Syoj · 2022-10-27

**Confidence:** 3
**Correctness:** 3
**Technical Novelty And Significance:** 2
**Empirical Novelty And Significance:** 2
**Recommendation:** 6

**Clarity, Quality, Novelty And Reproducibility:**

The paper is well-motivated, and the details provided in the paper could reproduce the experiments.



**Strength And Weaknesses:**

Strength:
(a) The author hypothesized that for crossmodal KD, distillation performance depends on the proportion of modality-general decisive features preserved in the teacher network, this hypothesis is refreshing to me.
(b) The verification experiments in Figure 2 and 3 are convincing to me.
Weaknesses:
(a) In some cases,  modality general decisive features and modality-specific decisive features of audio and video modalities could be imbalanced, e.g. on VGGSound Event dataset, there could be more audio decisive features than visual features feature, How would the proposed method handle this?


**Summary Of The Paper:**

The paper explores the topics of crossmodal knowledge distillation (KD), to transfer knowledge across modalities. To facilitate better understanding of crossmodal KD, the paper proposed a hypothesis that modality-general decisive features are the
crucial factor that determines the efficacy of crossmodal KD.

**Summary Of The Review:**

 The paper is overall well-motivated and the hypothesis is refreshing to me, however, I have some questions listed in the weaknesses part,

---

> ### Author Response · Authors · 2022-11-16
> **Responses to Reviewer Syoj**
>
> We thank Reviewer Syoj for the positive comments and for providing thoughtful feedback on our work. Below please see our detailed responses.
>
> **Imbalanced modality-specific features**
>
> We agree that in most real-world scenarios, modality-specific decisive features in the two modalities are imbalanced. Therefore, in our modality Venn diagram, we use two different notations, i.e., $\alpha$ and $\beta$, to represent the ratio of decisive features in modality $a$ and $b$ over all decisive features, respectively. In the VGGSound Event dataset mentioned by Reviewer Syoj, the fact of more audio decisive features than visual decisive features will translate to $\alpha\neq\beta$, or more specifically, $0 \leq \beta \leq \alpha \leq 1$ , if audio and video are taken as modality $a$ and $b$, respectively. Nevertheless, we want to clarify that our experiments are about varying $\gamma$ (i.e., $1-\alpha-\beta$) to support the proposed modality focusing hypothesis. The fact that $\alpha$ and $\beta$ correspond to different values won’t affect our designed approaches and experimental analysis.
>
> *In fact, our experiments have already taken modality-imbalance (i.e., $\alpha\neq\beta$) situations into consideration.* Taking our Gaussian synthetic experiment as an example, Figure 2 illustrates the scenario of $\alpha=\beta$ and Figure 3 discusses $\alpha \neq \beta$. We can see from Figure 3 that in the process of varying $\alpha$, $\alpha$ increases from 0 to 0.8,  while $\beta$ remains 0, so $\alpha$ and $\beta$ could be different along varying $\alpha$. To improve clarity and better address the reviewer’s concern, we have updated Figure 2 and 3 to include values of $\alpha$, $\beta$ and $\gamma$, and the data generation descriptions in Appendix B.2. For real-world multimodal data, we agree with Reviewer Syoj that $\alpha$ and $\beta$ are very likely to be different, and our algorithm is still applicable: modality imbalance doesn’t affect our analysis. In fact, for VGGSound experiments, we consider both settings: (1) teacher modality is audio and student modality is video; (2) teacher modality is video and student modality is audio. As can be seen from Table 4, our results are consistent for both settings: preserving modality-general decisive information in the teacher improves regular KD while modality-specific KD leads to performance degradation.
>
> To summarize, we reiterate that our proposed modality focusing hypothesis identifies the important role of $\gamma$, so $\alpha\neq\beta$ won’t impact our approaches, experiments, nor conclusions.
>
> We again thank Reviewer Syoj for reviewing our manuscript, and we hope that the above responses adequately address all concerns.

---

### Author Response · Authors · 2022-11-16
**Responses to All Reviewers**

We thank all reviewers for their thoughtful and constructive review of our manuscript. We were encouraged to hear that the reviewers found our proposed modality focusing hypothesis well-motivated (Reviewer Syoj, vsTN), refreshing (Reviewer Syoj) and significant (Reviewer i93D); and that they view our experiments reasonable (Reviewer vsTN) and rich (Reviewer i93D) to support the hypothesis.

We have uploaded a new version of the manuscript (revision highlighted by blue) incorporating reviewers’ comments. In summary, we: (1) updated Figure 2 & 3 and data generation details in Appendix B.2 to improve clarity; (2) added a new subsection (Sec. 2.3) to discuss related work; (3) explained differences of the modality Venn diagram with the multi-view assumption in Appendix C.

We would again like to thank all reviewers for their time and feedback, and we hope that our changes adequately address all concerns. Any further questions are highly welcomed. Below, we will provide individual responses to address each reviewer’s concerns.

&nbsp;

[1] Y. Tsai et al., Self-supervised Learning from a Multi-view Perspective, ICLR'21

[2] Y. Lin et al., Dual Contrastive Prediction for Incomplete Multi-View Representation Learning, TPAMI'22

[3] Y. Lin et al., COMPLETER: Incomplete Multi-view Clustering via Contrastive Prediction, CVPR'21

[4] Sridharan et al., An Information Theoretic Framework for Multi-view Learning, 2008

[5] Z. Xue et al., Multimodal Knowledge Expansion, ICCV’21

[6] J. Wang et al., Learning common and specific features for rgb-d semantic segmentation with deconvolutional networks, ECCV’16

[7] D. Hazarika, et al., Misa: Modality-invariant and-specific representations for multimodal sentiment analysis, Multimedia’20

[8] F. Ma et al., Learning better representations for audio-visual emotion recognition with common information, Applied Sciences’20.

[9] X. Sun et al., TCGM: An information-theoretic framework for semi-supervised multi-modality learning, ECCV’20.

---

### Decision · Program_Chairs · 2023-01-20

**Decision:**

Accept: notable-top-5%

**Justification For Why Not Higher Score:**

One reviewer has concerned about the suitable modality-general decisive features in real applications, which has not been well studied in this paper.

**Justification For Why Not Lower Score:**

The paper provides a thorough investigation of crossmodal knowledge transfer and proposed a hypothesis based on the investigations. Experimental results have confirmed the proposed hypothesis, diagnosed failure cases, and pointed directions to improve crossmodal knowledge transfer in the future.

**Metareview: Summary, Strengths And Weaknesses:**

1.  The paper provides a thorough investigation of crossmodal knowledge transfer and proposed a hypothesis based on the investigations. Experimental results have confirmed the proposed hypothesis, diagnosed failure cases, and pointed directions to improve crossmodal knowledge transfer in the future.
2. The motivation of this paper is clear, and the theoretical and experimental analyses are reasonable and easy to follow.
3. Future works are suggested to identify the most suitable modality-general decisive features in real applications.

**Note From Pc:**

if the above contains the word "oral" or "spotlight" please see: "oral" presentation means -> notable-top-5% and "spotlight" means -> notable-top-25%. As stated in our emails, we are disassociating presentation type from AC recommendations

**Summary Of Ac-Reviewer Meeting:**

The scores from the reviewers are quite consistent, and thus there was no AC-reviewer meeting for this paper.